# Inkjet-Printed Top-Gate Thin-Film Transistors Based on InGaSnO Semiconductor Layer with Improved Etching Resistance

**Siting Chen** [1], **Yuzhi Li** [1], **Yilong Lin** [1], **Penghui He** [1], **Teng Long** [1], **Caihao Deng** [1], **Zhuo Chen** [1], **Geshuang Chen** [1], **Hong Tao** [1,2,*], **Linfeng Lan** [1,*] and **Junbiao Peng** [1]

[1]  State Key Laboratory of Luminescent Materials and Devices, South China University of Technology, Guangzhou 510640, China

[2]  Guangzhou New Vision Optoelectronic Co., Ltd., Guangzhou 510530, China

*   Correspondence: th@newvision-cn.com (H.T.); lanlinfeng@scut.edu.cn (L.L.)

**Abstract:** Inkjet-printed top-gate metal oxide (MO) thin-film transistors (TFTs) with InGaSnO semiconductor layer and carbon-free aqueous gate dielectric ink are demonstrated. It is found that the InGaO semiconductor layer without Sn doping is seriously damaged after printing aqueous gate dielectric ink onto it. By doping Sn into InGaO, the acid resistance is enhanced. As a result, the printed InGaSnO semiconductor layer is almost not affected during printing the following gate dielectric layer. The TFTs based on the InGaSnO semiconductor layer exhibit higher mobility, less hysteresis, and better stability compared to those based on InGaO semiconductor layer. To the best of our knowledge, it is for the first time to investigate the interface chemical corrosivity of inkjet-printed MO-TFTs. It paves a way to overcome the solvent etching problems for the printed TFTs.

**Keywords:** inkjet printing; InGaSnO; thin-film transistors; top gate; metal oxide

## 1. Introduction

Over the past decade, great progress has been made in the development of the metal oxide (MO) thin-film transistors (TFTs). MO-TFTs based on vacuum processes have been applied to the commercialized display products such as liquid-crystal displays (LCDs) and active-matrix organic light-emitting displays (AMOLEDs), but their cost is relatively high due to the expensive vacuum systems and the complicated photolithography processes. In contrary, MO-TFTs fabricated by solution-based techniques are more attractive for their low fabrication costs and high throughput [1–4]. Among all of the solution-based techniques, inkjet printing, as the state-of-the-art drop-on-demand technique, is widely used in a variety of materials such as organic compounds, graphene, carbon nanotubes and oxides [5–8] and is particularly attracted in MO-TFT fabrication for its low material waste and high efficiency [9–11].

For inkjet-printed MO-TFTs, top-gate structure is more attractive compared to bottom-gate structure, because the gate dielectric layer can protect the channel from being affected by the air molecules or other defects introduced by the following processes [12]. Nevertheless, it is difficult to realize high-performance top-gate MO-TFTs via solution-processing method, because most of the amorphous MO semiconductors are sensitive to acidic environments [13]. That means amorphous MO semiconductors are easily affected by the process of the gate dielectric layer, leading to poor controllability and reproducibility [14]. It has been reported that indium tungsten oxide (IWO) has certain acid resistance prepared by vacuum process and spin-coating [15–18]. Obviously, inkjet-printing has the advantage of lower cost and greater potential of large-scale production.

In this paper, inkjet-printed top-gate MO-TFTs with a Sn-doped InGaO (InGaSnO) semiconductor layer were demonstrated. By doping Sn into InGaO, the chemical corrosivity is reduced. As a result, the printed InGaSnO semiconductor layer is almost not affected during printing the following gate dielectric layer. Although vacuum-processed InGaSnO films with the advantage of excellent electrical and optical properties has been reported before [19–26], it is for the first time to investigate the chemical corrosivity of inkjet-printed InGaSnO semiconductor films, to the best of our knowledge.

## 2. Experiment

### 2.1. Precursor Solutions

CYTOP solution was prepared by mixing solute (Asahi Glass, CTL-107MK) and solvent (Asahi Glass, CT-SOLV180) with volume ratio of 1:5 at room temperature with magnetic stirring at 700 rpm for 12 h. The InGaSnO precursor ink was prepared by dissolving $In(NO_3)_3 \cdot xH_2O$ (Aldrich, 99.99%, 0.14 M), $Ga(NO_3)_3 \cdot xH_2O$ (Aldrich, 99.99%, 0.02 M) and $SnCl_2 \cdot xH_2O$ (Aldrich, ≥99.995%, 0.04 M) in the mixture solvent of 2-methoxyethanol (Shanghai Aladdin Bio-Chem. Technology Co., Shanghai, China, 99.8%) and ethylene glycol (Shanghai Aladdin Bio-Chem. Technology Co., ≥99%) with a volume ratio of 1/1. For comparison, InGaO precursor solution was made by dissolving $In(NO_3)_3 \cdot xH_2O$ (0.18 M) and $Ga(NO_3)_3 \cdot xH_2O$ (0.02 M) in the mixture solvent. For $AlO_x$ precursor ink $Al(NO_3)_3 \cdot xH_2O$ (Aldrich, 99.99%, 0.4 M) was dissolved in deionized water. Both InGaSnO precursor ink and $AlO_x$ precursor ink were magnetically stirred at 700 rpm at room temperature for 12 h. Indium tin oxide (ITO) precursor solution was prepared by dissolving $In(NO_3)_3 \cdot xH_2O$ (0.475 M) and $SnCl_2 \cdot xH_2O$ (0.025 M) in the mixture solvent of 2-methoxyethanol and ethylene glycol with a volume ratio of 1/1. The ITO precursor ink was placed on a hot table at 50 °C and magnetically stirred at 700 rpm for 12 h. The $AlO_x$ precursor ink and CYTOP pure solvent were filtered by 0.22 μm syringe filter, and the other solutions were filtered by 0.45 μm syringe filter.

### 2.2. Device Fabrication

The surface-energy pattern and oxide film were fabricated with Dimatix (DMP-2850) inkjet printer (FUJIFILM Dimatix, Inc., Santa Clara, CA, USA) with a 10-pL ink cartridge. The glass substrate (Eagle XG, Corning, New York, NY, USA, 3 cm × 3 cm) is prepared by ultrasonic cleaning with water and isopropanol for 10 min respectively before deposition. The surface-energy pattern-assisted inkjet printing method was employed to prepare the aforementioned inkjet printing films [27], where the details of processing parameters are summarized in Table 1. The first step was to spin CYTOP solution onto the substrate at a speed of 3000 rpm for 40 s to generate a CYTOP layer with a thickness of 6 nm. The second step was to etch the CYTOP layer by printing pure CYTOP solvent (CT-SOLV180) (Asahi Glass, Tokyo, Japan) to form the desired surface-energy pattern. Then the patterned films were treated with oxygen plasma and ultraviolet radiation to remove unwanted residues and improve the wettability. So the oxide precursor ink could be printed into the surface-energy pattern. The schematic structure of the top-gate MO-TFTs is shown in Figure 1a. Firstly, a layer of 45 nm ITO (90:10 mol. % of $In_2O_3$:$SnO_2$) film was deposited on the glass substrate by DC magnetron sputtering (Beijing Technol Science Co., Ltd., Beijing, China, JCP-350) in a 100% Ar atmosphere. The base pressure was ~$8 \times 10^{-4}$ Pa, and the Ar gas flow was set to 9 sccm with the working pressure of 0.50 Pa. And then the pattern was visualized by photolithography to form pixel electrode, source/drain (S/D) electrodes, data lines, and $V_{dd}$ lines (Figure 1b). The InGaSnO layer was prepared by inkjet printing (Figure 1c). The channel and S/D region were then covered by $AlO_x$ films deposited by inkjet printing (Figure 1d). Then, the ITO film was inkjet printed to form the gate electrodes (Figure 1e). The gate electrode of driving TFTs is electrically connected to the source electrode of switching TFTs to form a driving circuit. Each of the printed oxide precursor film was then lightly baked for 5 min at 100 °C and hard baked for 1 h at 350 °C in the air, three times in total. The step profiles of different layers are shown in Figure S1.

**Table 1.** Processing Parameters of TFT Fabrication.

| Function Layer | Cytop Drop Spacing (μm) | Precursor ink Drop Spacing (μm) | Annealing Temperature (°C) | Annealing Time (h) |
|---|---|---|---|---|
| InGaSnO | 50 | 20 | 350 | 1 |
| AlO$_x$ | 10 | 5 | 350 | 1 |
| ITO gate | 40 | 25 | 350 | 1 |

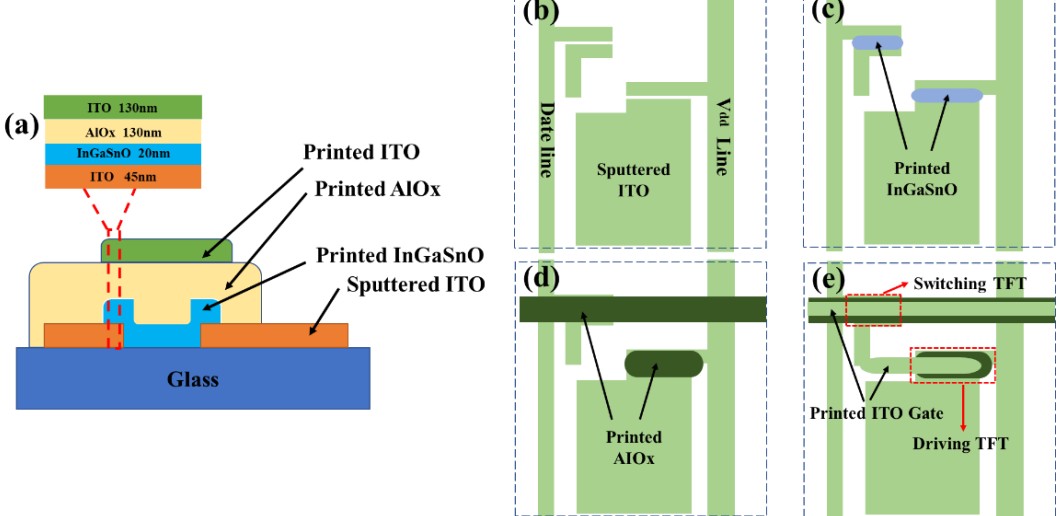

**Figure 1.** (**a**) Structural diagram of the printed top-gate MO TFTs. The fabrication flow for inkjet-printed MO TFTs: (**b**) preparing ITO electrodes by sputtering and photolithography. (**c**) Inkjet-printing MO semiconductor layers of the switching and driving TFTs. (**d**) Inkjet-printing AlO$_x$ dielectric films. (**e**) Inkjet-printing ITO gate electrodes.

### 2.3. Characterizations of Oxide Films and Devices

Nikon Eclipse E600 POL (Nikon Instruments, Inc., New York, NY, USA) was used for polarizing microscope image acquisition. The semiconductor films were characterized by X-ray diffraction (XRD, Bruker D8 ADVANCE, Karlsruhe, Germany). The chemical composition of the semiconductor film was characterized by X-ray photoelectron spectroscopy (XPS, ESCALAB 250Xi and Thermo Scientific). The electrical characteristics of TFTs were studied by using a semiconductor parameter analyzer (Keysight B1500A, Keysight Technologies, Inc., Santa Rosa, CA, USA). The film thickness was determined by a Daktak step profiler (Veeco Instruments, Inc., New York, NY, USA).

## 3. Results and Discussion

To improve the film quality of the AlO$_x$ dielectric layer, aqueous Al(NO$_3$)$_3$·xH$_2$O solution is chosen for the inks of the gate dielectric layer, because there are no carbon residues in the printed films based on the aqueous inks. Meanwhile, the water molecules are smaller compared with the molecules of organic solvents, so water molecules are easy to penetrate the oxide films without leaving nanopores or pinholes in the film during annealing. In contrary, the organic solvent will be evaporated or decomposed during annealing, leading to formation of nanopores, pinholes, and carbon residue impurities in the film [28]. However, the acidic environment of the aqueous AlO$_x$ precursor ink will corrode the MO semiconductor layer underneath (see Figure 2a, the InGaO semiconductor layer is obscure after AlO$_x$ gate dielectric film was printed on it).

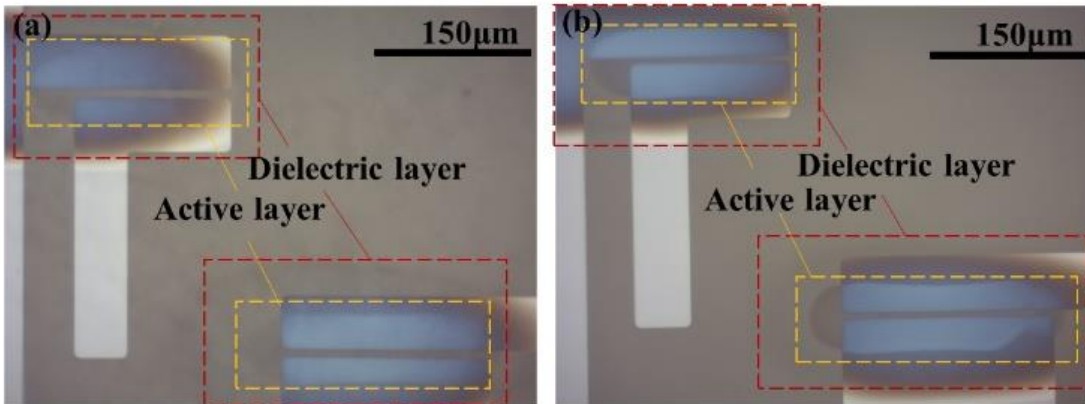

**Figure 2.** Polarizing microscope images of TFTs with (**a**) InGaO and (**b**) InGaSnO semiconductor layers.

To enhance the acid resistance of the MO semiconductors, Sn is doped into the InGaO semiconductors. To investigate the acid resistance of InGaO film and InGaSnO film to the same aqueous alumina precursor solution, InGaO and InGaSnO films with the same thickness were printed onto glass substrates under the same condition. When the two kinds of films are immersed in the same aqueous alumina precursor solution, the etching rate of the printed InGaSnO films at room temperature is ~0.05 nm/min, which is half of that of the printed InGaO (~0.1 nm/min). It proves our assumption that doping Sn into InGaO would increase acid resistance. Although the etching rates of both of InGaO and InGaSnO at room temperature are low, they increase greatly when the temperature increases to 100 °C (see Table 2). As mentioned in the experiment section, the printed aqueous $AlO_x$ precursor film experiences 100 °C soft drying for 5 min to evaporate the solvents. Therefore, testing the etching rates at 100 °C is close to the actual printing process of the MO TFTs. After doping Sn into InGaO, the etching rate at 100 °C reduces from 10 nm/min to 3 nm/min. The results show that InGaSnO are more compatible with the following printing process of the $AlO_x$ gate dielectric layer than InGaO.

**Table 2.** Etching Rate of the InGaO and InGaSnO Films in Aqueous $Al(NO_3)_3 \cdot xH_2O$ Solution.

| Temperature | Active Layer | Etching Rate |
|---|---|---|
| Room temperature | InGaO | 0.1 nm/min |
| | InGaSnO | 0.05 nm/min |
| 100 °C | InGaO | 10 nm/min |
| | InGaSnO | 3 nm/min |

Figure 2a,b shows the microscope images of the devices with the $AlO_x$ dielectric layer printing onto InGaO and InGaSnO, respectively. It shows that the InGaO film is corroded seriously by the following printing process of the $AlO_x$ gate dielectric layer (the InGaO film is blurred). By contrast, the InGaSnO film are seen clearly, indicating that the InGaSnO film has good etching resistance to the following printing process of the $AlO_x$ gate dielectric layer. In printed TFTs, solvent etching problem will lead to increase of interface defects and decrease of the electrical performance.

Figure 3 shows the X-ray diffraction (XRD) patterns of the InGaO and InGaSnO films. There is a weak peak (corresponding to nanocrystalline structure) appeared at approximately 32.0° for InGaO film, which is quite close to the (222) peaks in the $In_2O_3$ bixbyite structure crystals (about 30.6°) [29–31]. But the peak almost disappears for the InGaSnO film, indicating that doping Sn into InGaO leads to amorphous state. It is because that $SnO_2$ is generally in tetragonal rutile structure, which is different from that of the $In_2O_3$ and $Ga_2O_3$ (bixbyite structure). The fact that $Sn^{4+}$ ions substitute for $In^{3+}$ ones in crystal lattice reduce crystallinity. The decrease in crystalline quality is also probably caused by an aggregation of excess Sn element [32]. The increase in the acid resistance after doping Sn into InGaO

is attributed to the higher acid resistance of $SnO_2$. The presence of tin oxide improves the corrosion resistance of the entire InGaSnO semiconductor layer [33,34].

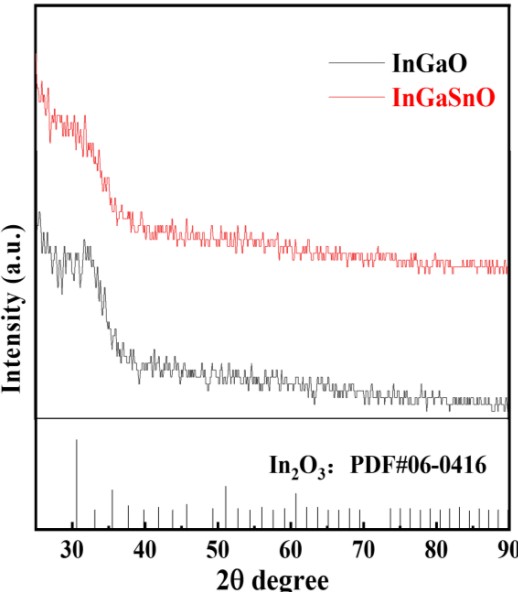

**Figure 3.** XRD spectra of the InGaO and InGaSnO films annealed at 350 °C for 1 h.

The chemical states of the printed InGaO and InGaSnO films were characterized by the XPS experiments. Figure 4a,b shows the XPS O 1s spectra for the InGaO and InGaSnO, respectively. Generally, the O 1s peaks can be fitted by three Gaussian distributions with binding energies of 529.8 eV, 530.9 eV and 531.6 eV, which are related to the oxygen in oxide lattices (M–O), oxygen vacancies ($V_O$) and metal hydroxide species (M–OH), respectively [35]. But the fitted peak near 531.6 eV is also related to the Sn–O, which is the reason for the apparent increase in the intensity of the fitted peak near 531.6 eV. Although the intensity of the fitted peak for $V_O$ (530.9 eV) decreases after doping Sn into InGaO, the intensity of the fitted peak for M–O (529.8 eV) also decreases. Therefore, it can be deduced that some of the Sn elements are not incorporated into InGaO lattice, leaving some $SnO_2$ segregated in the film (corresponding to 495.0 eV in Sn 3d3/2 spectrum, see the inset of Figure 4) [36]. Figures S2 and S3 shows the scanning electron microscope (SEM, Zeiss Merlin) images and the corresponding element distribution maps obtained from energy-dispersion x-ray spectroscopy (EDS) for InGaO and InGaSnO, respectively. It can be seen that element distributions are not uniform, which may be resulted from the defects formed by the chemical decomposition during annealing.

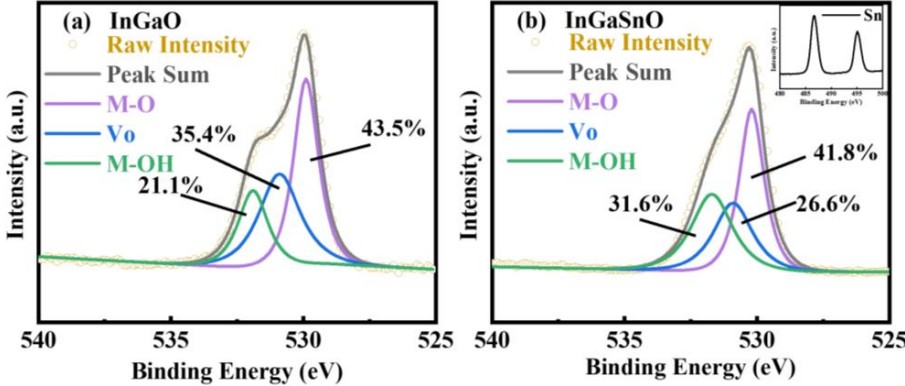

**Figure 4.** XPS O 1s spectra collected from (**a**) InGaO and (**b**) InGaSnO semiconductor layers. Inset: Sn 3d spectrum collected from InGaSnO semiconductor.

Figure 5 shows the output and transfer characteristics of the printed top-gate InGaO and InGaSnO TFTs. In the output curves, there are no current crowding effects in the linear regime, implying Ohmic contact between the MO semiconductor layer and the S/D electrodes (despite the bottom-contact structure). It is also observed that the output current of the InGaO TFT is much lower and the turn-on voltage ($V_{on}$) is much higher than that of the InGaSnO TFT, which is attributed to the serious damage of the InGaO film by the corrosive aqueous $AlO_x$ precursor ink. The mobility in the saturation region ($\mu_{sat}$) is calculated using the following

$$I_{DS} = \frac{W \mu_{sat} C_i}{2L}(V_G - V_T)^2, \tag{1}$$

where, $C_i$ is the areal capacitance of the dielectric (54.1 nF/cm$^2$ for the $AlO_x$ dielectric), $V_{th}$ is the threshold voltage, $L$ is the channel length (10 µm) and $W$ is the channel width (200 µm). The mobility of the InGaSnO TFT is ~3.0 cm$^2$V$^{-1}$s$^{-1}$, which is relatively high in printed MO TFTs, while the mobility of the InGaO TFT is only ~1.0 cm$^2$V$^{-1}$s$^{-1}$. Generally, the mobility of the printed MO TFTs is lower than that of the spin-coated ones, because the morphology and density of the printed MO films are worse than those of the spin-coated ones. The on/off current ratio, the threshold voltage and the subthreshold swing of the InGaSnO TFT are $1.59 \times 10^7$, −0.51 V, and 0.21 V /dec, respectively. For comparison, the corresponding data of InGaO TFT are $1.46 \times 10^9$, 5.46 V and 0.11 V /dec, respectively. The reason for the increase of threshold voltage may be that the InGaO film is etched, resulting in the film thickness reduction. Theoretically, doping Sn into InGaO would provide one electron carrier. However, the highly positive threshold voltage of InGaO TFT should be mainly attributed to the serious thickness reduction, because the InGaO TFT with bottom gate structure (without thickness reduction) is normally on (negative threshold voltage) [27].

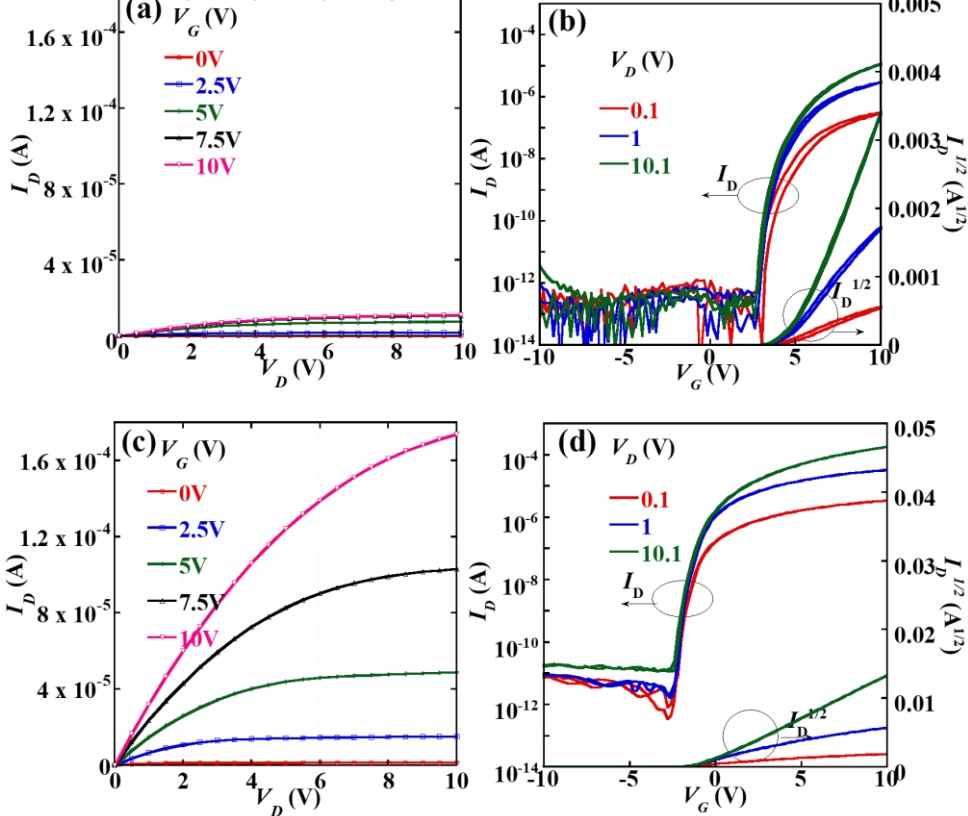

**Figure 5.** (**a**) Output and (**b**) transfer characteristics for the printed InGaO TFT; (**c**) output and (**b**) transfer characteristics for the printed InGaSnO TFT.

The electrical stability is important for the practical application of the MO-TFTs. It can be seen from Figure 5d that the hysteresis between forward and reverse sweeps of the transfer curves is negligible for the InGaSnO TFT, reflecting few fast electron traps in the InGaSnO film or at the InGaSnO/AlO$_x$ interface. To further study stability of the InGaSnO TFTs, the $V_{on}$ shift ($\Delta V_{on}$) under bias stress was measured at room temperature. $V_G$ was kept at 10 V for positive-bias stress (PBS) and −10 V for negative-bias stress (NBS) for 3600 s, and the transfer curves were recorded when $V_D = 10.1$ V was applied to InGaSnO TFT every 600 s. Figure 6a,b show the transfer curves of printed InGaSnO TFTs under PBS and NBS, respectively. $\Delta V_{on}$ for the device under PBS and NBS is 0.9 V and 0.1 V, respectively. In MO TFTs, NBS (at room temperature) stability is usually good, because holes can be hardly formed in the MO semiconductors when biased negatively. The relatively large $\Delta V_{on}$ for the InGaSnO TFT under PBS may be ascribed to some slow electron traps generated during printing the AlO$_x$ film onto the InGaSnO film. Although the PBS stability of the InGaSnO TFT is not good compared to those of vacuum-based MO-TFTs, it has been improved greatly compared to other vacuum-based InGaSnO TFT [37].

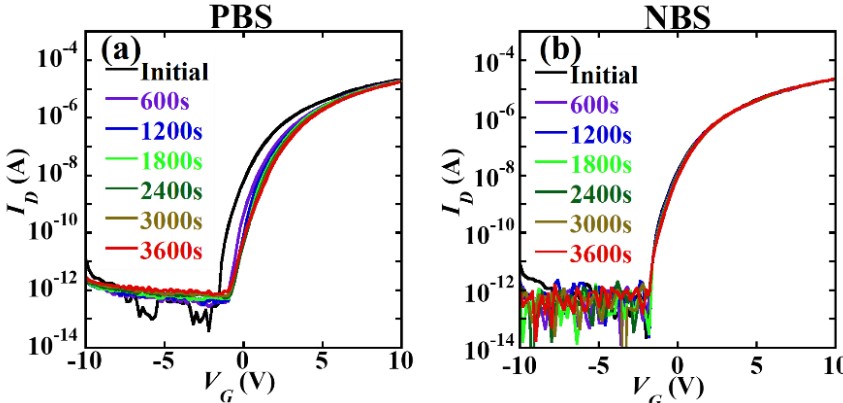

**Figure 6.** The variations of the time-dependent transfer curves of the InGaSnO TFTs under (**a**) PBS and (**b**) NBS.

## 4. Conclusions

In conclusion, inkjet-printed top-gate MO TFTs with a Sn-doped InGaO (InGaSnO) semiconductor layer were demonstrated. To improve the film quality of the AlO$_x$ dielectric layer, the aqueous Al(NO$_3$)$_3$·xH$_2$O solution is chosen for the inks of the gate dielectric layer, because there are no carbon residues in the printed films based on the aqueous inks. It is found that the etching resistance to the aqueous Al(NO$_3$)$_3$·xH$_2$O solution is enhanced by doping Sn into InGaO. As a result, the printed InGaSnO semiconductor layer is almost not affected during printing the following gate dielectric layer with an aqueous AlO$_x$ precursor ink. The TFTs based on the InGaSnO semiconductor layer exhibit higher mobility, and less hysteresis compared to those based on InGaO semiconductor layer and show excellent electrical stability. The study paves a way to overcome the solvent etching problems for the printed TFTs.

**Supplementary Materials:** The following are available online at http://www.mdpi.com/2079-6412/10/4/425/s1, Figure S1: The Daktak step profiling lines for (a) ITO S/D, (b) InGaSnO semiconductor, (c) AlOx insulator, and (d) ITO gate, Figure S2: SEM image and the energy-dispersion x-ray spectroscopy (EDS) element distribution maps of the InGaO film, Figure S3: SEM image and the EDS element distribution maps of the InGaSnO film.

**Author Contributions:** Conceptualization, L.L., Y.L. (Yuzhi Li), S.C., and P.H.; Formal Analysis, Y.L. (Yilong Lin), P.H., and Y.L. (Yuzhi Li); Investigation, C.D., Z.C., T.L., and J.P.; Writing—Original Draft, S.C. and G.C., Writing—Review Editing, L.L., H.T., and S.C. All authors have read and agreed to the published version of the manuscript.

**Funding:** This work was supported by the National Key R&D Program of China (Grant No. 2016YFB0401105), the National Natural Science Foundation of China (Grant No. 51673068), the Guangdong Natural Science Foundation (Grant No. 2017A030306007), the Guangdong Project of R&D Plan in Key Areas (Grant No. 2019B010934001), the

Guangdong Province Science and Technology Plan (Grant No. 2018A050506022), and the Zhujiang new science star project (Grant No. 201806010090).

**Acknowledgments:** The authors want to thank Yuzhi Li for his research contribution.

**Conflicts of Interest:** The authors declare no conflict of interest.

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
