# Peer review of "Inkjet-Printed Top-Gate Thin-Film Transistors Based on InGaSnO Semiconductor Layer with Improved Etching Resistance"

_coatings, doi:10.3390/coatings10040425_

Round 1
Reviewer 1 Report
Thanks to the authors for interesting work. This work is beneficial for the development of low cost TFT arrays. But some key information and evidences are missing, which are needed to include for a quality manuscript.
I have few comments and suggestions to the authors:
- Please add the information of the chemicals grade, purchased company etc information in the experiments section.
- I have a simple question, you didn't make any logic circuit this time or you didn't show any related data then why did you use so complicated structure or design of TFT? There are mistake in figure 1, page - 3. 1(C) definitely wrong and need to be corrected. and from 1(e) looks like gate electrode of one device is connected to S/D of another device. Is it true? How did you connect and measure? need clear explanation.
- From the figure 5 it is not clear about the devices performances. You should use same scale to have clear idea. From the present figure looks like InGaSnO have higher leakage current and higher contact resistance. But according to your data it is not matching.
- Similarly Figure 5(d) and figure 6 is not matching or can not be compared due to different scale.
- I would like to request the authors to evaluate the InGaO and InGaSnO semiconductor quality by evaluating channel resistance as well as contact resistance with ITO using TLM method.
Reviewer 2 Report
To improve the manuscript, please provide the following information and additional results:
- in "2.1. Precursor Solutions": the stirrer type and stirring rate should be specified for the CYTOP solution and precursor inks (InGaSnO, AlOx, and ITO);
- for the substrate should be specified the type or the trade name (supplier), surface roughness and size, and how the substrate was prepared/cleaned before the deposition;
- the results (prove) for thickness and 3D surface profiling measurements obtained using Daktak step profiler should be provided; the surface roughness of the films should be also provided;
- the oxide precursor inks need to be characterized in terms of stability, viscosity, surface tension, and pH value;
- for DC sputtering should be specified the magnetron type and deposition conditions for the ITO film on the glass substrate;
- for the XRD analysis should be specified in "2.3. Characterizations of Oxide Films and Devices" the type of the diffractometer and the measurement conditions, and in "3. Results and discussion", the ICDD reference card for In2O3 bixbyite structure should be specified;
- the chemical composition of InGaO and InGaSnO films should be specified;
- the morphology of InGaO and InGaSnO films should be analyzed by SEM.
Reviewer 3 Report
In this paper, the authors fabricate and characterize inkjet-printed top-gate metal-oxide thin film transistors (TFTs) with undoped or Sn-doped InGaO semiconductor layers. They demonstrate that the chemical corrosivity of InGaO is reduced by Sn doping. Hence, InGaSnO layers enable top-gate TFTs with better performance that their InGaO counterparts.
The paper is timely and interesting. The results are worthy of publication. The experimental work is well described and analyzed. All conclusions are based on experimental evidence.
The paper can be ready for publication after a minor revision.
Here are comments and suggestions for the authors:
There are a few English mistakes to correct. A couple of examples: “In contrary, MO-TFTs fabricated by solution-based techniques are more attractive for its low fabrication costs and high throughput [1-4].” Its -> their; “As mentioned in the experiment section, the printed the aqueous AlOx precursor film experiences…”, repeated the … the, etc.
“ It has been reported that IWO has certain acid resistance …” Here and elsewhere define the acronyms.
Figure 1c): The two arrows point to the inkjet-printed MO semiconductor layers; the “sputtered ITO” label should be overwritten.
“The reason for the increase of threshold voltage may be that the InGaO film is etched, resulting in the film thickness thinning.” I am not sure that this is a good reason for the high threshold voltage. Try to give a deeper explanation. What about the doping effect of Sn?
Round 2
Reviewer 1 Report
Thanks to the authors for their to the point response.
Reviewer 2 Report
The manuscript is recommended for publishing since the authors answered satisfactorily to the raised questions.